# Resveratrol and Female Fertility: A Systematic Review

**DOI:** 10.3390/ijms252312792

**Published:** 2024-11-28

**Authors:** Alessandro Bertoldo, Damiano Pizzol, Dong Keon Yon, Maura Callegari, Valentina Gobbo, Pierluigi Cuccurese, Laurie Butler, Susanna Caminada, Justin Stebbing, Fiona Richardson, Julia Gawronska, Lee Smith

**Affiliations:** 1U.O.S.D. of Assisted Reproductive Technologies “G. Beltrame”, Ospedale di Oderzo, ULSS2, 31046 Treviso, Italy; alessandro.bertoldo@gmail.com (A.B.); maura.callegari1@aulss2.veneto.it (M.C.); valentina.gobbo@aulss2.veneto.it (V.G.); 2Health Unit, Eni, Maputo 1100, Mozambique; 3Center for Digital Health, Medical Science Research Institute, Kyung Hee University Medical Center, Kyung Hee University College of Medicine, Seoul 02447, Republic of Korea; yonkkang@gmail.com; 4Department of Obstetrics and Gynecology, Ospedale di Oderzo, ULSS2, 31046 Treviso, Italy; pierluigi.cuccurese@aulss2.veneto.it; 5Faculty of Science and Engineering, Anglia Ruskin University, Cambridge CB1 1PT, UK; laurie.butler@aru.ac.uk; 6Health Unit, Eni, 00144 Rome, Italy; susanna.caminada@eni.com; 7Department of Life Sciences, Anglia Ruskin University, Cambridge CB1 1PT, UK; justin.stebbing@aru.ac.uk; 8The Queen Elizabeth Hospital King’s Lynn NHS Foundation Trust, King’s Lynn PE30 4ET, UK; fiona.richardson@qehkl.nhs.uk; 9Centre for Health, Performance and Wellbeing, Anglia Ruskin University, Cambridge CB1 1PT, UK; julia.gawronska@aru.ac.uk (J.G.); lee.smith@aru.ac.uk (L.S.)

**Keywords:** resveratrol, 3,5,4-trihydroxystilbene, female fertility, female infertility

## Abstract

Resveratrol is a natural polyphenolic compound that may have multiple influences on human health, including antiaging, anti-inflammatory, anti-neoplastic, antioxidant, insulin-sensitizing, cardioprotective and vasodilating activities. Growing evidence also suggests a potential positive effect of resveratrol on female fertility. The aim of the present study was to collate and appraise the scientific literature on the relationship between resveratrol and female fertility. We systematically searched Medline, PubMed, Web of Science and Embase from the databases’ inception (1951, 1951, 1947 and 1900, respectively) until 9th May 2024. All in vivo or in vitro retrospective or prospective studies reporting the effects of resveratrol interventions on women’s fertility were included. We ultimately incorporated twenty-four studies into a systematic review with a narrative summary of the results; of those studies, nine were performed on women seeking natural or assisted fertility, and fifteen were in vitro studies performed on human cells and tissues in different stages of the reproductive cascade. The current literature, though limited, suggests that resveratrol may play a role in female infertility. Specifically, it may significantly and positively impact reproductive outcomes, owing to its potential therapeutic effects improving ovarian function. Further studies are now needed to better understand resveratrol’s effects and define the optimal dosage and periods of intake to maximize beneficial effects, as well as to prevent adverse outcomes on implantation, subsequent pregnancy and the fetus.

## 1. Introduction

Resveratrol (3,5,4-trihydroxystilbene) is a natural polyphenolic compound present in a variety of plants, foods and drinks [1]. In plants, it is produced in response to ultraviolet radiation, injury, fungal or bacterial infection and it is predominantly found in the skin of grapes, blueberries, raspberries, mulberries and peanuts [2]. Resveratrol may have many effects on human health, including antiaging, anti-inflammatory, anti-neoplastic, antioxidant, insulin-sensitizing, cardioprotective and vasodilating activities [3]. In particular, this antiaging activity is carried out through resveratrol’s ability to improve cellular mitochondrial activity and trigger a series of molecular mediators capable of counteracting some of the most important metabolic mechanisms of aging [4,5,6]. Moreover, resveratrol exhibits antioxidant properties by scavenging free radicals and enhancing the activity of antioxidant enzymes. Additionally, it exerts anti-inflammatory effects through a variety of signaling pathways inhibiting pro-inflammatory cytokines [7]. Growing evidence suggests a potential positive effect of resveratrol on the infertile population, particularly females, by potentially enhancing ovarian function, improving oocyte quality and exerting protective effects against age-related fertility decline and polycystic ovary syndrome (PCOS) [8]. In particular, in vitro, resveratrol inhibits proliferation and androgen production by theca–interstitial cells, exerting a cytostatic but not cytotoxic effect on granulosa cells, while decreasing aromatization and vascular endothelial growth factor expression [8]. Interestingly, in vivo, resveratrol treatment has been found to reduce the size of adipocytes and improve estrus cyclicity in an acyclic rat model of polycystic ovary syndrome (PCOS), also increasing the ovarian follicular reserve and prolonging the ovarian life span [9]. Another study found that after resveratrol treatment, mitochondrial membrane potential and ATP content in oocytes of aging mice was increased, resulting in the restoration of oocyte quality without adverse effects in the animals or their offspring [10]. At low doses, resveratrol activates SIRT1, an NADH-dependent deacetylase, able to program cellular energy metabolism through transcriptional regulation of the pGC-1α gene, a master regulator of mitochondrial metabolic activity, suggesting the ability of resveratrol to optimize such activity [11]. SIRT-1 can activate LKB1 by deacetylation; LKB1 phosphorylates and activates AMP-dependent kinase (AMPK) in a reciprocal positive feedback relationship. Indeed, activated AMPK itself can increase the concentration of NAD+ by promoting further activation of SIRT-1. Both AMPK and SIRT-1 are involved in the activation of PGC-1α, which is responsible for the co-activation of nuclear respiratory factors (NRF-1 and NRF-2), which induce transcription of genes involved in mitochondrial biogenesis and mitochondrial transcription factor A (TFAM). This process is directly involved in mitochondrial DNA replication [11]. Ragonese et al. confirmed in vitro and ex vivo the mitochondrial activity of resveratrol on granulosa cells, showing that activation of SIRT-1 and AMPK by resveratrol promotes increased mitochondrial membrane potential, ATP production and mitogenesis. This suggests the impact of resveratrol as an energy enhancer for granulosa cells, which are essential for oocyte development and maturation [12].

Interestingly, a limited body of literature also suggests a role of resveratrol in enhancing male fertility by improving testicular function [13] and sperm quality [14]. However, there is no consensus on the usage and posology of resveratrol. Moreover, literature on the relationship between resveratrol and female fertility is contrasting. For these reasons, the aim of this systematic review was to explore the impact of resveratrol on female fertility, examining data extracted from both in vivo and in vitro studies performed in females and human cells or tissues.

## 2. Methods

This systematic review with a narrative summary of the results adhered to the PRISMA [15] and MOOSE [16] statements and followed a structured protocol available under reasonable request from the corresponding author.

### 2.1. Search Strategy

Two investigators (AB and DP) independently conducted a literature search using the Medline, PubMed, Web of Science and Embase databases from their inception (1951, 1951, 1947 and 1900, respectively) to the 9th of May 2024. The following search strategy was used: “(3,5,4′-Trihydroxystilbene” OR “3,4′,5-Stilbenetriol” OR “trans-Resveratrol-3-O-sulfate” OR “SRT 501” OR “cis-Resveratrol” OR “Resveratrol” OR “trans-Resveratrol” OR “Resveratrol-3-sulfate”) AND (“oocytes” OR “oocytes development competency” OR “fertilization” OR “blastulation” OR “pregnancy” OR “live birth” OR “oocyte quality” OR “embryo” OR “embryo quality” OR “in vitro fertilization” OR “fertilisation in vitro” OR “IVF” OR “ICSI” OR “assisted reproduct*” OR “reproduct* medic*”). The references of the retrieved articles together with the proceedings of relevant conferences were hand-searched to identify other potentially eligible studies for inclusion in the analysis missed by the initial search or any unpublished data. The literature search, assessment of inclusion and exclusion criteria, quality assessment of studies and extraction of data were independently undertaken and verified by two investigators (AB, DP). The results were then compared, and in the case of a discrepancy, a consensus was reached with the involvement of a third senior investigator (LS). There was no language restriction applied.

### 2.2. Types of Studies; Inclusion and Exclusion Criteria

Studies included in the present review had to meet the following criteria: (1) the study design was retrospective, cross-sectional, prospective, a randomized clinical trial (RCT) or another experimental design; (2) the population included women seeking fertility; (3) resveratrol supplementation was included as the intervention or an exposure group; (4) the study contained a female control group not treated with resveratrol supplementation; (5) the study reported effects of resveratrol on women’s fertility.

Studies were excluded if they had a study design different from those stated in the inclusion criteria, focused on males, contained no data on resveratrol supplementation or did not investigate any aspect of fertility.

### 2.3. Data Extraction and Statistical Analyses

For each eligible study, two independent investigators (AB, DP) extracted the following data: name of the first author and year of publication, article type, study design, sample size, sample characteristics, intervention, outcome measures and findings.

### 2.4. Outcomes

The primary outcome was the effect of resveratrol supplementation on female fertility. Secondary outcomes included in vitro findings related to the effect of resveratrol on fertility cascade processes.

### 2.5. Assessment of Study Quality

For human studies two independent authors (DP, AB) assessed study quality using the Newcastle–Ottawa Scale (NOS) [17]. The NOS assigns a maximum of 9 points based on three quality parameters: selection, comparability and outcome. As per the NOS grading in past reviews, we graded studies as having a high (<5 stars), moderate (5–7 stars) or low risk of bias (≥8 stars) [18]. Notably, for in vitro studies, there is no general consensus and standard tool for assessing study quality.

### 2.6. Assessment of the Certainty of Evidence

To ascertain the certainty of the evidence, the Grading of Recommendations Assessment, Development, and Evaluations (GRADE) framework was used [19].

## 3. Results

As shown in Figure 1, we initially found 395 possibly eligible articles. After removing 366 papers through title/abstract screening, 29 were retrieved as full text. Of the twenty-nine full-text articles, five studies were excluded—four because they focused on male fertility and one because it assessed resveratrol levels without supplementation—leaving twenty-four studies published between 2010 and 2024 to be included in the systematic review [12,20,21,22,23,24,25,26,27,28,29,30,31,32,33,34,35,36,37,38,39,40,41,42]. Among these, nine were performed on women seeking natural or assisted fertility [20,21,22,23,24,25,26,27,28], and fifteen were in vitro studies performed on human cells and tissues in different stages of the reproductive cascade [12,29,30,31,32,33,34,35,36,37,38,39,40,41,42]. Most of the studies (n = 10) were conducted in Europe, nine in Asia, three in South America, one in the Middle East and one in Oceania. The nine in vivo studies included a total of 9563 participants, and descriptions of the characteristics of these studies and their main findings are reported in Table 1. The main results are varied and not always consistent. Resveratrol induced a reduction in the expression of the vascular endothelial growth factor and hypoxia-inducible factor 1 genes in the granulosa cells. The number of mature oocytes, cleavage rate, fertilization rate and fertility rate were not significantly different, but the high-quality oocyte rate and high-quality embryo rate were higher in the resveratrol group [20]. It also induced modification in the miRNome reflecting transcriptomic and proteomic modification in granulosa cells. The number of fertilized good-quality oocytes increased in treated women, and a significant anticorrelation between miR-125 fold change values and biochemical pregnancy was present [21]. Resveratrol treatment was associated with statistically significant increases in the follicle output rate and follicle-to-oocyte index with no difference in the number of oocytes retrieved, biochemical pregnancy, clinical pregnancy or live birth rates [22]. The time needed to control blood pressure in resveratrol-treated women was significantly reduced, while time before a new crisis was extended. The number of treatment doses needed to control blood pressure was lower in treated women. No differences in maternal or neonatal adverse effects were observed between the two groups [23]. Again, resveratrol supplementation was associated with significantly higher numbers of oocytes and MII oocytes retrieved, a higher fertilization rate, more cleavage embryos per patient, more blastocytes per patient and more cryopreserved embryos. No significant differences in biochemical or clinical pregnancy, live birth or miscarriage rates were revealed, but a trend toward a higher live birth rate was revealed in the resveratrol group [24]. Resveratrol treatment promoted remodeling of the scarred uterus, regeneration of the endometrium and muscular cells and vascularization. It also improved the pregnancy rate compared with patients receiving placebo [25]. No difference in systolic or diastolic parameters between treated and control-group obese women were reported. All blood chemistry parameters improved compared to placebo at 30 days and significantly improved at 60 days with respect to placebo. Resveratrol also significantly improved lipid and glucose parameters at 30 to 60 days of treatment [26]. No difference in the treatment of pain in endometriosis was observed [27]. Resveratrol intake was also reported to be strongly associated with a decrease in the clinical pregnancy rate and an increased risk of miscarriage [28].

Table 2 reports the characteristics and the main findings of the in vitro studies.

No publication bias test was performed. The median quality of the studies performed on human studies was 5.9 (range: 4–9), indicating an overall satisfactory quality. Because of the high heterogeneity, the certainly of this evidence has been rated as moderate.

## 4. Discussion

The findings of our systematic review suggest a general positive impact of resveratrol on female reproductive health. Although the wide range of aspects considered in the included studies does not allow robust conclusions, the results do suggest a potential positive effect of resveratrol on multiple domains of the female reproductive system. The number and quality of matured oocytes were investigated by four and two studies, respectively. The number was reported as being increased in two [21,24] and having no significant differences in the others [20,22]. Both studies assessing quality reported an increased quality in women who underwent resveratrol supplementation. A possible explanation is the direct action of resveratrol in reducing oxidative stress, protecting mitochondrial DNA from damage and mutations, while enhancing telomerase activity and reducing cellular aging. Moreover, it activates sirtuin 1 (SIRT1), a key molecule in aging, which is typically reduced in aged oocytes, making them vulnerable to oxidative stress [43]. By compensating for this decreased SIRT1 expression, resveratrol may inhibit ovarian aging and extend ovarian lifespan [8,44]. Additionally, it potentially exerts positive effects on PCOS and obesity-related infertility by inhibiting pathways involved in androgen production and reducing inflammation and oxidative stress [45,46]. Moreover, in a rodent premature ovarian insufficiency model induced by chemotherapy or radiotherapy, resveratrol inhibited oxidative stress and inflammatory events in the ovaries by activating the PI3K/Akt/mTOR and NF-κB signaling pathways [47], improving loss of the oogonial stem cells through antiapoptotic effects [48]. In in vitro fertilization treatments, it was observed that resveratrol enhanced oocyte maturation and embryo development to the blastocyst stage in both animals and humans [31] and protected against postovulatory oocyte aging [49]. Fertilization and fertility rates were reported as being improved by Gerli et al. [24] and as having no difference by Bahramrezaie and colleagues [20]. The latter study also reported no difference in cleavage rate but found an increased embryo quality [20]. The follicle output rate and follicle-to-oocyte index were reported as improved in one study [22], and two studies reported no difference in terms of live birth [22,24]. Importantly, there are contrasting data regarding pregnancy rates and miscarriage indicators. The pregnancy rate was reported by five studies, with two indicating an increase [21,25], two no difference [22,24] and Ochiai et al. reporting a decrease [28]. Ochiai and colleagues also reported an increase in the miscarriage rate [28], while Gerli et al. observed no difference [24]. Moreover, during spontaneous pregnancies in overweight patients, resveratrol was found to significantly improve lipid and glucose parameters [26], and the literature suggests that it may be beneficial for the treatment of preeclampsia in combination with oral nifedipine [23]. This contrasting evidence could be explained by the fact that, since implantation requires an inflammatory response with local secretion of proinflammatory cytokines and prostaglandins from the decidualized endometrium, the anti-inflammatory properties of resveratrol might directly suppress embryo implantation. In turn, this could imply an optimization of the supplementation scheme in terms of period, dosage and duration. Finally, no effect in pain relief in women suffering endometriosis was observed [27], although in vitro studies showed antiapoptotic and antiproliferative effects with a possible role in inhibiting the progression of ectopic endometrium [9]. Thus, resveratrol may have therapeutic benefits for treating infertility associated with endometriosis, even without symptomatic effects. Taken together, all these findings do not allow putative conclusions to be drawn, but they spark optimism for the possible use of resveratrol for women seeking fertility. Many questions still need to be addressed, including the dosage, treatment duration, optimal time window and possible side and teratogenic effects. However, additional encouraging results are coming from in vitro studies that also shed light on possible mechanisms of action. Importantly, in vitro studies included in the present review assessed different aspects of the impact of resveratrol on human cells, indicating an overall potential beneficial effect of supplementation. Interestingly, its antioxidant activity was reported by multiple studies [29,30,34,39,40]. Granulosa cells’ viability was also found to be improved by supplementation [12,34,38], and other observed effects included an anti-inflammatory response [30]; oocyte maturation [31]; myometrium relaxation [35]; reduction of dehydroepiandrosterone, androstenedione and 11-deoxicortisol [37]; and neuroprotection [41]. Contrasting results were reported on decidualization. Three studies showed enhancement of in vitro decidualization [32,33,42], while Ochiai and colleagues reported anti-deciduogenic properties [36]. In this case, different methodological approaches, resveratrol doses and timing of administration may explain the contrasting results, highlighting the need to increase efforts in studying this multi-potential supplement. Despite some contrasting evidence, it seems clear that resveratrol may play an important role in female fertility management. Moreover, a limited body of literature also suggests a role of resveratrol in improving testicular function and sperm quality through enhanced protection from reactive oxygen species (ROS), which negatively impacts sperm quality by damaging mitochondrial membranes, impairing sperm motility and increasing sperm DNA damage [13,14].

The considerations drawn in this systemic review should be interpreted in the light of some limitations including the limited number of in vivo studies; the different parameters taken into account by each study and the lack of robust data on dosage, side effects and teratogenic effects. Moreover, although the quality of the studies was satisfactory overall, due to the high heterogeneity, the certainly of this evidence has been rated as moderate. Finally, the present review was not registered in the PROSPERO database.

Resveratrol is generally considered safe and well tolerated when consumed in moderate amounts through diet and in supplemental doses up to 5 g/day for a month [50,51]; however, the safety of high-dose supplementation, particularly over long periods, remains unclear. Potential side effects include gastrointestinal disturbances and interactions with medications. Therefore, it is crucial to conduct further and larger studies to determine the safe and effective dosage of resveratrol for improving female fertility. Considering its limited absorption after oral administration, particular attention should also be paid to resveratrol’s bioavailability and, in particular, to the different resveratrol-based technologies that can ameliorate resveratrol’s pharmacokinetic characteristics, exploiting its biopharmaceutical potential [52].

## 5. Conclusions

In conclusion, although it is not possible to define conclusive indications on resveratrol supplementation, the current evidence suggests that its utilization for women seeking fertility and during pregnancy could significantly and positively impact reproductive outcomes, particularly because of its potential therapeutic effects in improving ovarian function. Further studies are needed to better understand resveratrol’s potential in women in order to define the optimal doses and periods of intake to maximize beneficial effects and to prevent adverse effects on implantation, subsequent pregnancy and the fetus.

## Figures and Tables

**Figure 1 ijms-25-12792-f001:**
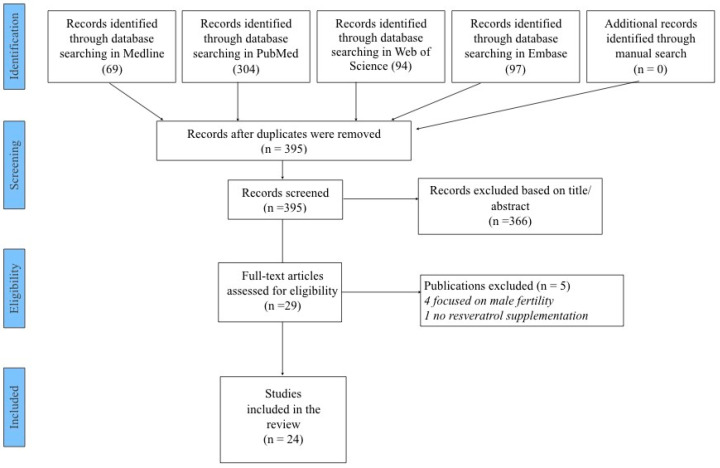
PRISMA flowchart.

**Table 1 ijms-25-12792-t001:** Main information and findings of included studies on fertility and reproductive outcomes.

Author and Date	Aim	Study Type	Intervention	Outcome Measures	Findings
Ding, 2017 [23]	To evaluate the outcome of treatment combining oral nifedipine and resveratrol against preeclampsia	RCT	400 women with preeclampsia	Time to control blood pressure and time before a new hypertensive crisis, number of doses needed to control blood pressure and maternal and neonatal adverse effects	In the resveratrol group, the number of treatments and time needed to control blood pressure were significantly reduced, while time before a new crisis was extended.
Malvasi, 2017 [26]	To investigate the effect of trans-resveratrol during spontaneous pregnancies in overweight patients	RCT	110 pregnant women aged 25–40 years, gestational age at enrollment between the 24th and 28th weeks and BMI in first trimester between 25 and 30	Blood pressure, total cholesterol, LDL, HDL, triglycerides and blood glucose	All blood chemistry parameters improved compared to placebo at 30 days and significantly at 60 days with respect to placebo. The resveratrol group showed significantly improved lipid and glucose parameters compared to the DC/MI group after 30 to 60 days of treatment.
Mendes da Silva, 2017 [27]	To evaluate resveratrol utilization for reducing endometriosis pain	RCT	44 women (ages 20–50) with laparoscopic diagnosis of endometriosis	Pain assessment using a visual analog scale	Resveratrol is not superior to placebo for treatment of pain in endometriosis.
Ma, 2018 [25]	To investigate the effects of resveratrol in patients with a scarred uterus	Cohort study	78 patients (mean age 30.4) with a scarred uterus, randomly divided into resveratrol treatment (n = 46) and placebo (n = 32) m groups	Uterus scar modeling and fertility	Resveratrol treatment promoted remodeling of the scarred uterus, regeneration of the endometrium and muscular cells and vascularization and improved the pregnancy rate.
Bahramrezaie, 2019 [20]	To assess the effect of resveratrol on the angiogenesis pathway for management of PCOS	RCT	62 ICSI candidates with PCOS	VEGF and HIF1 gene expression, number and quality of mature oocytes and embryos, cleavage rate, fertilization rate and fertility rate	There was a reduction in the expression of the VEGF and HIF1 genes under the effect of resveratrol in the granulosa cells. The high-quality oocyte rate and high-quality embryo rate were higher in the resveratrol group.
Ochiai, 2019 [28]	To assess resveratrol’s impact on IVF–embryo transfer	Cohort study	8686 embryo transfers	Pregnancy outcomes	Resveratrol intake is strongly associated with a decrease in clinical pregnancy rate and an increased risk of miscarriage.
Gerli, 2021 [24]	To evaluate the impact of resveratrol on ICSI	RCT	101 infertile women undergoing ICSI, aged 18–42, BMI 18–30, normal thyroid function and normal blood parameters, regular uterine cavity	Number of developed follicles, total oocytes, MII oocytes recovered, fertilization rate, number of cleavage embryos/blastocysts, number of embryos for cryopreservation, duration and dosage of gonadotropins, number of embryos per transfer, implantation, pregnancy rates, live birth rate and miscarriage rate	Resveratrol supplementation was associated with significantly higher numbers of oocytes and MII oocytes retrieved, a higher fertilization rate, more cleavage embryos/patient, more blastocytes/patient, more cryopreserved embryos and a higher live birth rate.
Battaglia, 2022 [21]	To evaluate follicular fluid miRNome modification in aged women with a poor ovarian reserve receiving a resveratrol-based supplement for 3 months	Cohort study	12 women 35–42 years old with a poor ovarian reserve (AMH < 1.2 ng/mL, AFC < 5) undergoing IVF treatment	MiRNome modifications and oocytes quality	The number of fertilized good-quality oocytes increased in treated women, and a significant anticorrelation between miR-125 fold change values and biochemical pregnancy was present.
Conforti, 2024 [22]	To evaluate the effect of resveratrol on the outcome of IVF	RCT	70 women >35 years with good ovarian reserve (AMH > 1.2 ng/mL)	Follicle output rate, follicle-to-oocyte index, number of oocytes retrieved, biochemical pregnancy, clinical pregnancy and live birth rates	Resveratrol treatment was associated with a statistically significant increase in the follicle output rate and follicle-to-oocyte index.

AFC = antral follicle count; AMH = anti-Müllerian hormone; DC = D-chiro-inositol; HIF = hypoxia-inducible factor; ICSI = intracytoplasmic sperm injection; IVF = in vitro fertilization; MI = myo-inositol; PCOS = polycystic ovary syndrome; RCT = randomized clinical trial; VEGF = vascular endothelial growth factor.

**Table 2 ijms-25-12792-t002:** Main information and findings of included studies on in vitro outcomes.

Author and Date	Aim	Sample Size and Characteristics	Intervention	Outcome Measures	Findings
Schube, 2010 [38]	To evaluate the effect of resveratrol against oxLDL-induced damage to granulosa cells	Granulosa cells obtained from patients undergoing in vitro fertilization	Cells were treated with 150 μg/mL oxLDL alone or with 30 μM resveratrol for 36 h	Measurement of oxidative stress markers, cell vitality and activity	Resveratrol protected granulosa cells by reducing cell death; enhancing mitosis; inducing protective autophagy; reducing oxidative stress markers and reducing expression of LOX-1, TLR4, CD36 and heat-shock protein 60. Resveratrol could restore steroid biosynthesis.
Novaković, 2015 [35]	To evaluate the in vitro effect of resveratrol on the oxytocin-induced contractions of term pregnant myometrium and the contribution of different K^+^ channels to resveratrol’s action	Myometrial samples from 42 nonlaboring women undergoing elective cesarean section in the third trimester of pregnancy, mean age 35.46 years	Resveratrol was dissolved in 70% *v/v* ethanol with further dilution in distilled water before use; working concentrations of ethanol in the bath were <0.01% (*v*/*v*)	Levels of oxytocin-induced contractions of myometrium cells and K^+^ channel activity	Resveratrol induced a concentration-dependent relaxation of myometrium contractions. The inhibitory effect of low-concentration resveratrol involves different myometrial K^+^ channels. When applied in high concentrations, resveratrol has an additional K^+^-channel-independent mechanism(s) of action.
Savchuk, 2016 [37]	To characterize the effects of resveratrol on human fetal adrenal steroidogenesis	Primary cultures of human fetal adrenocortical cells prepared from adrenals of aborted fetuses (GW10–12)	Fetal adrenocortical cells were treated in the presence or absence of ACTH (10 ng/mL) with or without resveratrol (10 μM) for 24 h	Dehydroepiandrosterone, androstenedione and 11-deoxicortisol levels; activity of cytochromes 17αhydroxylase/17,20 lyase and 21-hydroxylase	Resveratrol significantly suppressed synthesis of dehydroepiandrosterone, androstenedione and 11-deoxicortisol by ACTH-activated and unstimulated human fetal adrenocortical cells, which was associated with inhibition of the activity and expression of cytochromes 17αhydroxylase/17,20 lyase and 21-hydroxylase.
Hannan, 2017 [30]	To assess resveratrol’s anti-inflammatory and anti-oxidative effects in trophoblast and endothelial cells	NR	Trophoblasts were treated with 0–100 μM resveratrol for 48 h in culture; HUVECs were treated with 0–75 μM resveratrol for 24 h in culture	Measurement of sFlt-1 and sEng or protein expression of peNOS, eNOS or HO-1	Resveratrol reduced sFlt-1, sFlt-1 e15a and soluble endoglin secretion from trophoblasts and HUVECs and reduced mRNA expression of the pro-inflammatory molecules NFκB, IL-6 and IL-1β in trophoblasts. IL-6, IL-1β and TNFα secretion were also significantly reduced. In HUVECs, resveratrol significantly increased mRNA of the antioxidant enzymes HO-1, NQO1, GCLC and TXN but did not significantly alter HO-1 protein expression, while it reduced HO-1 protein in trophoblasts.
Liu, 2018 [31]	To evaluate the effects of resveratrol on oocyte maturation in aged mice and humans	64 women 38–45 years of age undergoing ICSI	3 different concentrations of resveratrol (0.1, 1.0 and 10 mm) or dimethylsulfoxide	Oocyte maturation, fertilization, immunofluorescence intensity of mitochondria and normal morphology	Resveratrol at 1.0 mm significantly increased the first polar body emission rate in oocytes. The immunofluorescence intensity of mitochondria and normal morphology of spindle and chromosome of oocytes undergoing in vitro maturation were notably improved.
Caldeira-Dias, 2019 [29]	To investigate the effect of resveratrol on endothelial cells from women before the development of preeclampsia regarding antioxidant defenses and vasodilator factors	6 samples from women who developed severe preeclampsia and 6 samples from women who had healthy pregnancies	HUVECs were incubated in medium containing 10% (*v*/*v*) plasma from case and control patients and 1 μM trans-resveratrol	Levels of Nrf2, HO-1, GSR, GSH and NO in endothelial cells	Resveratrol prevents alterations in HO-1 and NO markers and improves GSH levels.
Ochiai, 2019 [36]	To assess the effect of resveratrol on HESC decidualization	Endometrial biopsies from patients without overt uterine pathology during the luteal phase	Confluent monolayers were maintained in DMEM/F12 without phenol red containing 2% (*v*/*v*) DCC-FBS and treated with 0.5 mM 8-bromo-cAMP and 1 μM P4 with or without 100 µM resveratrol	Expression levels of prolactin and IGFBP1; cell decidualization	Resveratrol has anti-deciduogenic properties, repressing the induction of the decidual marker genes prolactin and IGFBP1 but also abrogating decidual senescence. Resveratrol blocks differentiation of HESCs into mature and senescent cells by accelerating downregulation of the CRABP2-RAR pathway.
Mestre Citrinovitz, 2020 [33]	To evaluate the effect of resveratrol on decidualization of HESCs	Endometrial biopsies from healthy, regularly cycling women mean age 34.4 years old	At days 3 and 5 of the decidualization, different doses of resveratrol (0 (vehicle treatment), 6.25, 12.5, 25 and 50 µM) were added	Expression levels of prolactin and IGFBP1, cell proliferation and mRNA levels	Resveratrol increased the expression levels of prolactin and IGFBP1, indicating enhanced in vitro decidualization of HESCs. It was accompanied by a decrease in cell proliferation and by changes in the mRNA levels of key cell cycle regulators.
Viana-Mattioli, 2020 [39]	To investigate the effects of trans-resveratrol on oxidative stress and NO production in women with preeclampsia, gestational hypertension and healthy pregnancies	10 blood samples collected for each group from non-smokers women < 34 years	Cells were incubated with 10 μM of trans-resveratrol	ROS production, SIRT1 activity and NO levels	In the gestational hypertension group, resveratrol decreased intracellular ROS and increased their antioxidant capacity, while inhibiting SIRT1 reestablished previous levels. In preeclampsia, inhibition of SIRT1 increased antioxidant activity. Intracellular NO and supernatant nitrite levels were increased by inhibiting SIRT1 in the preeclampsia group.
Wang, 2020 [41]	To investigate the protective effects of resveratrol against oxidative-stress-induced damage in trophoblasts	A human first-trimester extravillous trophoblast cell line purchased from a supplier	Cells were pre-treated with 12.5, 25, 50 and 100 μM resveratrol for 24 h, followed by 200 μM H_2_O_2_ for 12 h	ROS, MDA and SOD levels; cell viability and apoptosis and SIRT1 levels	Pre-treatment with resveratrol significantly ameliorated H_2_O_2_-induced cytotoxicity, morphological damage, oxidative stress and apoptosis.
Moreira-Pinto, 2021 [34]	To evaluate the direct effects of resveratrol on granulosa cell viability, steroidogenic function and oxidative stress	34 women undergoing assisted reproductive technology, mean age 34 years	Granulosa cells were treated with resveratrol (0.001–20 µM) for different lengths of time (24–72 h)	ROS/RNS levels and granulosa cell viability and steroidogenic function	Low concentrations of resveratrol suggest a protective role reducing ROS/RNS formation after inducement of stress. High concentrations of resveratrol affect granulosa cells’ viability and steroidogenic function.
Ragonese, 2021 [12]	To study the effects of resveratrol on the growth, electrophysiology and mitochondrial function of human granulosa cells	7 infertile women undergoing assisted reproductive techniques	Cells were treated with various concentrations of resveratrol (3, 10 and 20 mM) for up to 72 h	Granulosa cell viability and mitochondrial activity; electrophysiological activity of potassium current and calcium concentration	Resveratrol induced mitochondrial activity with a bell-shaped, dose-dependent relationship. It increased ATP production and cell viability and promoted the induction of cellular differentiation. Resveratrol reduced the functional expression of an ultra-rapidly activated, slowly inactivated, delayed rectifier potassium current that is associated with plasma membrane depolarization and that promotes an increase in intracellular calcium.
Yao, 2021 [42]	To examine resveratrol’s effect in rescuing defects caused by zearalenone in HESCs during human decidualization	NR	Unspecified resveratrol treatment	ROS levels and glutathione peroxidase 3 gene expression	Resveratrol restored the impaired decidualization process by induction of the anti-oxidative gene glutathione peroxidase 3.
Wang, 2022 [41]	To investigate resveratrol’s neuroprotective effect during development	Human induced pluripotent stem cells purchased from a supplier	Cells were treated with 2, 10 and 50 μM resveratrol	Cell proliferation, apoptosis and differentiation	Resveratrol showed neuroprotective effects by promoting neural cell proliferation, inhibiting apoptosis and accelerating the differentiation of germ layers.
Long, 2023 [32]	To examine resveratrol’s effects on defects caused by DEHP during human decidualization	NR	After DEHP treatment, cells were treated with resveratrol (RSV) at a cell density of 70%	Cell proliferation and decidualization and the up/downregulation of molecules associated with decidualization	Resveratrol treatment was associated with an upregulation of decidual molecules, confirmed by RNA-seq transcriptome analysis and a quantitative real-time PCR assay.

ACTH = adrenocorticotropic hormone; CRABP2 = cellular retinoic acid-binding protein 2; DEHP = Di-(2-Ethylhexyl) phthalate; DMEM = Dulbecco’s modified Eagle’s medium; GSH = glutathione; GSR = glutathione reductase; HESC = human endometrial stromal cell; HO-1 = heme oxygenase-1; HUVEC = human umbilical vein endothelial cell; ICSI = intracytoplasmic sperm injection; IGFBP1 = insulin-like growth factor binding protein 1; MDA = malondialdehyde; NO = nitric oxide; NR = not reported; Nrf2 = nuclear factor-erythroid-derived 2-related factor-2; oxLDL = oxidized low-density lipoprotein; ROS = reactive oxygen species; SOD = superoxide dismutase.

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
