# Peer review of "Resveratrol and Female Fertility: A Systematic Review"

_ijms, 2024, doi:10.3390/ijms252312792_

Round 1

Reviewer 1 Report

Comments and Suggestions for Authors

In this manuscript authors screened the current literature in order to find a relationship between resveratrol and female fertility. 

Although the manuscript is interesting and generally well written, there are several points that deserve to be improved. In particular:

Line 126: Results and discussion section must be divided

Table 1: Data reported in Findings column should be more concise  

An accurate revision of typing errors is recommended

Abbreviations must be written in full length when mentioned for the first time

References: Correct lines 266-267

It is surprising that such a simple review required 12 authors. A detailed authors contribution section is required 

Author Response

Dear Editor, 

Please find enclosed the revised version of the Manuscript ID ijms-3332521 entitled " Resveratrol and female fertility: a systematic review".

We carefully revised the manuscript considering all suggestions you provided us. 

Reviewer: 1
Although the manuscript is interesting and generally well written, there are several points that deserve to be improved. In particular:

Line 126: Results and discussion section must be divided

Thank you, we have now divided out the results and discussion as has been suggested.

Table 1: Data reported in Findings column should be more concise  

Thank you for your suggestion, we have now limited the content to ensure the findings are more concise.

An accurate revision of typing errors is recommended.

A complete revision and editing by a professor native in English has been carried out.

Abbreviations must be written in full length when mentioned for the first time

Thank you, as suggested we have now provided the full-length description for each abbreviation when first mentioned.

References: Correct lines 266-267

Thank you, done.

It is surprising that such a simple review required 12 authors. A detailed authors contribution section is required.

Thank you for your question. We do not feel that 12 authors are excessive, and it is fewer than most systematic reviews. To conduct a systematic review such as this requires a large team of researchers, not only experts in the methodological approach but also experts in relation to exposure and all the subsequent health outcomes, as well as methodological design of included studies. Moreover, the different geographical location of the authors represents an added value for a better comprehension of a novel matter. Multidisciplinary and multinational scientific teams should not be discouraged; they are essential for the advancement of science. 

Here the specific contribution:

  • Conception/design of the work; LS, DKY, MC
  • Methodology: JG, JS
  • Data acquisition AB, DP, JG, PC
  • Data analysis/interpretation: VG, LB, SC, FR
  • Drafting the work: SC, FR, JS
  • Critical revision: JS, LS, DKY
  • Supervision: LS, AB
  • Final approval of the version: ALL AUTHORS
  • Agreement to be accountable for all aspects of the work in ensuring that questions related to the accuracy or integrity of any part of the work are appropriately investigated and resolved: ALL AUTHORS

Reviewer 2 Report

Comments and Suggestions for Authors

The review titled “ijms-3332521_Resveratrol and female fertility: a systematic review” is submitted to the section “Bioactives and Nutraceuticals ” of the Special Issue “Potential Health Benefits of Resveratrol: Laboratory-Confirmed Mechanisms of Action ”.

The purpose of this systematic review was to compile and appraise the scientific literature on the relationship between resveratrol and female fertility, focusing on data from both “in vivo” and “in vitro” studies conducted in women and in human cells or tissues.

Regarding the abstract, it should indicate the four databases searched and the complete time frame of the review, not just the end date. A systematic review involves rigorously gathering, evaluating, and synthesising available evidence on a specific research question, using a structured methodology to assess potential biases. Tools such as GRADEor the Cochrane Risk of Bias Tool are commonly employed for this purpose.

The introduction provides an overview of resveratrol’s characteristics but does not sufficiently develop the research question regarding its potential positive effects on infertility, specifically in women. As this is the core of the work, it should be expanded with more detailed information. References, such as those grouped (8-10 and 11-13), should ideally be broken down to provide clarity. I suggest articulating the biological plausibility to clearly introduce the research question that leads to the study’s stated objective.

In the Materials and Methods section, the full review period should be specified, including both the start and end dates. For structuring the research question, the PICO framework should be used; however, the authors have included it in the criteria for inclusion/exclusion and study types in Section 2. I recommend revising and correcting this part. The study type is a systematic review, so it’s essential to define inclusion and exclusion criteria clearly, as these shape the selection of studies and, consequently, the findings. These criteria are not currently described in detail.

The authors assess methodological quality and risk of bias using the Newcastle-Ottawa Scale (NOS), which is specifically recommended for observational studies, i.e., case-control and cohort studies. However, it appears that other study designs are included in this review; please clarify this aspect. Additionally, quality is assessed using GRADE, which is recommended for systematic reviews. However, there is no information on structured data extraction or bias assessment.

Moreover, systematic reviews are often registered on platforms like PROSPERO; the authors should indicate whether this step was completed. The selection of articles should adhere to PRISMA guidelines, which should be presented in the methodology section rather than in the results, and the PRISMA diagram should include a separate box for each database identified.

The authors present results alongside the discussion. For Tables 1 and 2, studies are typically listed in chronological order from oldest to most recent. Additionally, the results column lacks numerical data, making it challenging to interpret; it should include results with confidence intervals to enhance clarity. The quality assessment indicated in the methodology section is also missing.

The results and discussion section is very brief; the results should be explained in greater detail, and the discussion should consider quality aspects specific to a systematic review, such as scores. Furthermore, the limitations of the studies included should be acknowledged, along with any biases, to help future research avoid these issues.

The conclusion does not fully serve its purpose, as there is no real assessment of findings following the review. I recommend rewriting this section comprehensively.

Author Response

Dear Editor, 

Please find enclosed the revised version of the Manuscript ID ijms-3332521 entitled " Resveratrol and female fertility: a systematic review".

We carefully revised the manuscript considering all suggestions you provided us. 

Reviewer: 2
Regarding the abstract, it should indicate the four databases searched and the complete time frame of the review, not just the end date. A systematic review involves rigorously gathering, evaluating, and synthesising available evidence on a specific research question, using a structured methodology to assess potential biases. Tools such as GRADE or the Cochrane Risk of Bias Tool are commonly employed for this purpose.

Thank you, as suggested, we included the four databases and the starting data. We decided to avoid any further information in the abstract in relation to the methods employed, owing to limitations in relation to word count and the requirement of appropriate information within the other sections. 

The introduction provides an overview of resveratrol’s characteristics but does not sufficiently develop the research question regarding its potential positive effects on infertility, specifically in women. As this is the core of the work, it should be expanded with more detailed information. References, such as those grouped (8-10 and 11-13), should ideally be broken down to provide clarity. I suggest articulating the biological plausibility to clearly introduce the research question that leads to the study’s stated objective.

Thank you, as you suggested we have now better articulated these points in the introduction.

In the Materials and Methods section, the full review period should be specified, including both the start and end dates. 

Thank you, but it is already stated “from inception to 9th of May” that is from database inception.

For structuring the research question, the PICO framework should be used; however, the authors have included it in the criteria for inclusion/exclusion and study types in Section 2. I recommend revising and correcting this part. 

Thank you for this suggestion, whilst our research question is structured based on PICO, in fact, it is most common and potentially more accurate to structure and describe in detail the inclusion/ exclusion criteria of a systematic review using PICO, as per journal guidelines and also multiple other sources, see for example, Cochrane Library About PICO | Cochrane Library for a comprehensive understanding of how PICO is used in systematic reviews. 

The study type is a systematic review, so it’s essential to define inclusion and exclusion criteria clearly, as these shape the selection of studies and, consequently, the findings. These criteria are not currently described in detail.

Thank you, the criteria are reported in section 2.2.Following the PICOS (participants, intervention, controls, outcomes, study design) criteria, we included studies assessing: P: Women seeking fertility I: Resveratrol supplementation C: Women not treated with resveratrol supplementation O: Effects of resveratrol on women fertility S: All retrospective, cross-sectional or prospective, randomized clinical trials (RCT) and experimental studies reporting the effects of resveratrol interventions on women fertility were included. Studies were excluded if they had no data on resveratrol supplementation or if no fertility aspect was investigated.”

The authors assess methodological quality and risk of bias using the Newcastle-Ottawa Scale (NOS), which is specifically recommended for observational studies, i.e., case-control and cohort studies. However, it appears that other study designs are included in this review; please clarify this aspect. 

Thank you, we specified that the NOS is used for human studies. For in vitro studies, although some adaptation proposal exists (https://discovery.ucl.ac.uk/id/eprint/10121267/3/Norris_Accepted%20version%20-%20Modified%20NOS.pdf), there is no general consensus and standard tool for assessing the quality so we did not apply.

Additionally, quality is assessed using GRADE, which is recommended for systematic reviews. However, there is no information on structured data extraction or bias assessment.

Thank you, in the results section we state that “Because of the high heterogeneity, the certainly of this evidence has been rated as moderate.” For more details on GRADE per se we already provide an appropriate reference. We included reference to GRADE in the limitations section to elucidate on this aspect.

Moreover, systematic reviews are often registered on platforms like PROSPERO; the authors should indicate whether this step was completed. The selection of articles should adhere to PRISMA guidelines, which should be presented in the methodology section rather than in the results, and the PRISMA diagram should include a separate box for each database identified

Thank you. In the methods section we reported the applied tool, and in results the output of the application. However, we have no objection if the editor prefers to move in other section. In the same section we specified that “a structured protocol is available under reasonable request from the corresponding author.”

We integrated figure 1, PRISMA diagram, with the additional data requested.

The authors present results alongside the discussion. For Tables 1 and 2, studies are typically listed in chronological order from oldest to most recent. Additionally, the results column lacks numerical data, making it challenging to interpret; it should include results with confidence intervals to enhance clarity.

Thank you, based on the other reviewer’s recommendation we had to further reduce the data present in tables…. Thus, addressing this point directly conflicts with the other reviewer, thus, we ask for the editor to inform us if he prefers the approach proposed in this comment.

The quality assessment indicated in the methodology section is also missing.

Thank you, we specified that the NOS was applied to human studies and we reported results. For in vitro studies, although some adaptation proposal exists (https://discovery.ucl.ac.uk/id/eprint/10121267/3/Norris_Accepted%20version%20-%20Modified%20NOS.pdf), there is no general consensus and standard tool for assessing the quality so we did not apply.

The results and discussion section is very brief; the results should be explained in greater detail, and the discussion should consider quality aspects specific to a systematic review, such as scores. Furthermore, the limitations of the studies included should be acknowledged, along with any biases, to help future research avoid these issues. The conclusion does not fully serve its purpose, as there is no real assessment of findings following the review. I recommend rewriting this section comprehensively.

Thank you, we have edited the discussion section addressing the suggested points. However, although the results section may seem short, it has been balanced with table content in order to not weight down the reading. Limitations have been updated with reference to the quality aspect. In our opinion conclusions are in line with the purpose, highlighting on one side the necessity of more and more numerous studies, and mentioning, on the other side, the most investigated aspect of female fertility in relation to resveratrol supplementation.

Reviewer 3 Report

Comments and Suggestions for Authors

This is an interesting analysis and would merit publication with suitable revisions.  I like the slant to tying reproductive health to mitochondrial function; this is a point that I believe has substantial and growing corroborative evidence.

The analysis seems a bit limited in one sense, however, in that it didn't mention the resveratrol has a Michael acceptor, and thus is prone to covalent adduction to proteins.  It is reasonable to wonder whether such an attribute could drive its pharmacological involvement and preferential targeting.  Such an understanding could help to rationalize trends evident within the meta-analysis.

A second point worth questioning is the absence of any discussions regarding FSH and luteinizing hormone, both of which play major roles in reproductive function and dysfunction, and both have also been discussed functionally with respect to resveratrol.

Comments on the Quality of English Language

There are numerous minor compositional glitches, including (but not limited to):

line 166:  "two studies reported no difference in term of"  >>  should be 'terms'

line 202:  "a limited body of literature also suggest a role"  >> should be 'suggests'

Lines 111-116:  The following statement:

"Please note that the publication of your manuscript 111 implicates that you must make all materials, data, computer code, and protocols 112 associated with the publication available to readers. Please disclose at the submission 113 stage any restrictions on the availability of materials or information. New methods and 114 protocols should be described in detail while well-established methods can be briefly 115 described and appropriately cited"

... is obviously out of place.  There is a slight change it relates to study inclusion criteria but, if so, it is substantially misphrased.  More likely, the authors copied publisher instructions and somehow pasted them into the manuscript.

Lines 117-124:  same problem as above.

Author Response

Dear Editor, 

Please find enclosed the revised version of the Manuscript ID ijms-3332521 entitled " Resveratrol and female fertility: a systematic review".

We carefully revised the manuscript considering all suggestions you provided us. 

Revewer 3

Comments and Suggestions for Authors

This is an interesting analysis and would merit publication with suitable revisions.  I like the slant to tying reproductive health to mitochondrial function; this is a point that I believe has substantial and growing corroborative evidence.

The analysis seems a bit limited in one sense, however, in that it didn't mention the resveratrol has a Michael acceptor, and thus is prone to covalent adduction to proteins.  It is reasonable to wonder whether such an attribute could drive its pharmacological involvement and preferential targeting.  Such an understanding could help to rationalize trends evident within the meta-analysis.

Thank you for your interesting and illuminating comment. As already stated, more and more numerous studies are necessary for a full understanding of this topic and for driving strength in conclusions. We would like not to include this aspect in the discussion as we excluded from the criteria pre-clinical and animal studies. Based on human and human cells and tissues studies, none of them are currently dealing with this aspect, so the discussion of which is out of scope for the present review.

A second point worth questioning is the absence of any discussions regarding FSH and luteinizing hormone, both of which play major roles in reproductive function and dysfunction, and both have also been discussed functionally with respect to resveratrol.

Thank you, also this, as previous point is so interesting and important. Unfortunately, as for the previous point, none of the studies included mentioned this aspect. 

Comments on the Quality of English Language

There are numerous minor compositional glitches, including (but not limited to):

 A complete revision and editing by a professor native in English has been carried out.

line 166:  "two studies reported no difference in term of"  >>  should be 'terms'

Thank you, edited.

line 202:  "a limited body of literature also suggest a role"  >> should be 'suggests'

 Thank you, edited.

Lines 111-116:  The following statement:

"Please note that the publication of your manuscript implicates that you must make all materials, data, computer code, and protocols associated with the publication available to readers. Please disclose at the submission stage any restrictions on the availability of materials or information. New methods and protocols should be described in detail while well-established methods can be briefly described and appropriately cited"

... is obviously out of place.  There is a slight change it relates to study inclusion criteria but, if so, it is substantially misphrased.  More likely, the authors copied publisher instructions and somehow pasted them into the manuscript.

Thank you, indeed this is a mistake when formatting the manuscript for the present journal.

Lines 117-124:  same problem as above.

Thank you, the same response as above.

Round 2

Reviewer 2 Report

Comments and Suggestions for Authors

1.    Thank you very much for allowing me to review the article titled “ijms-3332521_Resveratrol and female fertility: a systematic review” is submitted to the section “Bioactives and Nutraceuticals once again, as well as the authors’ responses to the suggestions provided.
2.    Regarding the abstract, as I previously mentioned, the time period covered by the review needs to be clearly specified, including both the starting and ending years, not just the conclusion of the timeframe.
3.    In relation to the introduction, I had requested an expanded explanation of the physiological basis through which resveratrol protects female fertility to strengthen the scientific foundation of the literature review. However, aspects related to male fertility have been included instead, which I believe is not particularly relevant to this review. Therefore, I propose that lines 62 to 67 be removed and replaced with content that outlines the physiological mechanisms supporting the biological plausibility of the protective effect. Additionally, I had suggested disaggregating the references (e.g., references 8 to 10 or 11 to 13) to provide more precise information, but this has not been addressed.
4.    In the methodology section, I had recommended specifying both the start and end dates of the review period, which has not been incorporated. Furthermore, as this is a systematic review, I recommend registering it in PROSPERO. I also noted that while the PICO framework is widely used in evidence-based medicine and clinical research to structure and clarify research questions, it does not replace inclusion and exclusion criteria. Therefore, the authors must include these criteria explicitly.
5.    The article selection flowchart (Figure 1, PRISMA) is part of the methodology, not the results, and should be moved to the methodology section. In my previous comments, I suggested that the PRISMA flowchart should include a separate box for each database in the identification stage. Although the authors’ letter indicates this change has been made, it is not reflected in the manuscript.
6.    As I mentioned in my earlier review, the results section is too brief, with most findings presented in the discussion section. This is not appropriate, as results should be included in the results section, while comments and interpretations belong in the discussion section. The authors cite disagreements with other reviewers regarding this issue; therefore, I suggest that the editor makes the final decision deemed most appropriate.
7.    Additionally, I believe the tables should be arranged chronologically, from the oldest articles to the most recent ones.
8.    Regarding the conclusion, this should directly address the objective of the review. The objective, as stated in the title, focuses on female fertility. However, the conclusion, as well as other parts of the manuscript, also discusses male fertility, which creates confusion throughout the paper. This issue should be clarified to maintain consistency.

Author Response

Dear Editor, 

Please find enclosed the revised version of the Manuscript ID ijms-3332521 R1 entitled " Resveratrol and female fertility: a systematic review".

We carefully revised the manuscript considering all suggestions you provided us.

Reviewer: 2
1.    Thank you very much for allowing me to review the article titled “ijms-3332521_Resveratrol and female fertility: a systematic review” is submitted to the section “Bioactives and Nutraceuticals once again, as well as the authors’ responses to the suggestions provided. 

Thank you for your hard work on reviewing our manuscript, we have addressed all your comments raised.

2.    Regarding the abstract, as I previously mentioned, the time period covered by the review needs to be clearly specified, including both the starting and ending years, not just the conclusion of the timeframe. 

Thank you for your comment. It is not clear to us as to exactly what you mean in terms of the search dates. Generally, when reporting the search for a systematic review one will state the dates searched in the databases that is database inception to 9th May 2024, which is correct for this review. It would be highly unusual to state the date of all databases inception and we are not aware of any other example of this in the academic literature. If reading explicitly the reviewer is asking for the start date and the end date of the review period, meaning the day we started the search to the day we downloaded the papers for screening, this is almost an instantaneous process and thus is done on the same day. If the reviewer is referring to the date the first paper was published then this information belongs in the results. We do apologise but the comment is not clear and we hold the reviewer in the highest regard, but, what we have stated is correct and we provide references here to systematic reviews our team have produced with this information, this is a sample from several hundred reviews produced by our team.  Urinary incontinence and quality of life: a systematic review and meta-analysis - PMC Prevalence of erectile dysfunction in male survivors of cancer: a systematic review and meta-analysis of cross-sectional studies - PMC Non‐pharmacological approaches for treatment of premature ejaculation: a systematic review - Pizzol - 2023 - Trends in Urology & Men's Health - Wiley Online Library

3.    In relation to the introduction, I had requested an expanded explanation of the physiological basis through which resveratrol protects female fertility to strengthen the scientific foundation of the literature review. However, aspects related to male fertility have been included instead, which I believe is not particularly relevant to this review. Therefore, I propose that lines 62 to 67 be removed and replaced with content that outlines the physiological mechanisms supporting the biological plausibility of the protective effect. Additionally, I had suggested disaggregating the references (e.g., references 8 to 10 or 11 to 13) to provide more precise information, but this has not been addressed.

We removed male integrations.

We added physiological basis.

We disaggregated references.

4.    In the methodology section, I had recommended specifying both the start and end dates of the review period, which has not been incorporated. Furthermore, as this is a systematic review, I recommend registering it in PROSPERO. I also noted that while the PICO framework is widely used in evidence-based medicine and clinical research to structure and clarify research questions, it does not replace inclusion and exclusion criteria. Therefore, the authors must include these criteria explicitly. 

Thank you for your comment. It is not clear to us as to exactly what you mean in terms of the search dates. Generally, when reporting the search for a systematic review one will state the dates searched in the databases that is database inception to 9th May 2024, which is correct for this review. It would be highly unusual to state the dates of inception of all the databases searched and we are not aware of any example of this in the academic literature. If reading explicitly the reviewer is asking for the start date and the end date of the review period, meaning the day we started the search to the day we downloaded the papers for screening, this is almost an instantaneous process and thus is done on the same day. If the reviewer is referring to the date the first paper was published then this information belongs in the results. We do apologise but the comment is not clear and we hold the reviewer in the highest regard, but, what we have stated is correct and we provide references here to systematic reviews our team have produced with this information, this is a sample from several hundred reviews produced by our team.  Urinary incontinence and quality of life: a systematic review and meta-analysis - PMC Prevalence of erectile dysfunction in male survivors of cancer: a systematic review and meta-analysis of cross-sectional studies - PMC Non‐pharmacological approaches for treatment of premature ejaculation: a systematic review - Pizzol - 2023 - Trends in Urology & Men's Health - Wiley Online Library

It is no longer possible to register in PROSPERO as the review is already completed, registration can only take place prior to the review or early in the review process.

 We have now removed PICO from the inclusion/exclusion criteria and have explicitly explained as suggested. 

5.    The article selection flowchart (Figure 1, PRISMA) is part of the methodology, not the results, and should be moved to the methodology section. In my previous comments, I suggested that the PRISMA flowchart should include a separate box for each database in the identification stage. Although the authors’ letter indicates this change has been made, it is not reflected in the manuscript.

Figure position: we placed the figure in results section as in our opinion is the most appropriate as it is after the first mention in the text. It is not uncommon practice to do this. However, we have no objection if the editor prefers to move into another section at the formatting stage. Often journals will have a standard guideline where to place a figure such as this which is executed at the formatting stage. 

Figure organization: in R1 we added the data regarding each database in the whole box leaving the previous format as, in our opinion clearer. We now provided a new format with the 4 boxes.

6.    As I mentioned in my earlier review, the results section is too brief, with most findings presented in the discussion section. This is not appropriate, as results should be included in the results section, while comments and interpretations belong in the discussion section. The authors cite disagreements with other reviewers regarding this issue; therefore, I suggest that the editor makes the final decision deemed most appropriate. 

Thank you, we edited results section adding main findings.

7.    Additionally, I believe the tables should be arranged chronologically, from the oldest articles to the most recent ones. 

As suggested we arranged chronologically.

8.    Regarding the conclusion, this should directly address the objective of the review. The objective, as stated in the title, focuses on female fertility. However, the conclusion, as well as other parts of the manuscript, also discusses male fertility, which creates confusion throughout the paper. This issue should be clarified to maintain consistency.

Thank you, as suggested we removed from conclusion male fertility issue focusing just on female aspects. 

Round 3

Reviewer 2 Report

Comments and Suggestions for Authors

I have carefully reviewed the new version of the manuscript “ijms-3332521_Resveratrol and female fertility: a systematic review” is submitted to the section “Bioactives and Nutraceuticals ” as well as the reviewers’ response to the suggestions aimed at improving the clarity of the work. I appreciate the effort made in enhancing this review study.
However, there are several key points that need to be addressed:
1.    Time period covered: Specifying the time period covered is essential in a review, as it helps contextualise the study and connect it with other related reviews. Similarly, the design of the review must be clearly defined.
2.    Thematic consistency: Discussing male infertility in a manuscript titled "Female Infertility" is inconsistent and detracts from the main focus of the work.
3.    Registration in PROSPERO: The lack of registration in PROSPERO is a notable limitation, particularly given that the review is already complete. Registration is a widely recognised standard that adds transparency and credibility to systematic reviews.
While improvements are evident, addressing these aspects is crucial to enhance the quality and coherence of the manuscript.

Author Response

Time period covered: Specifying the time period covered is essential in a review, as it helps contextualise the study and connect it with other related reviews. Similarly, the design of the review must be clearly defined.

Thank you for highlighting this again.  In our previous response we requested more clarity as to what exactly the review wants here. We are sorry but the same confusion continues, as the question remains the same, and it is not clear. We have already followed the standard reporting of the search window for systematic reviews. However, we have now also included in the methods  "from database inception  (1951, 1951, 1947, 1900 respectively). In the results section we have now stated studies were published between 2010 and 2024. 

We have also now clearly stated in the abstract and methods that this is a systematic review with a narrative summary of the results.

Thematic consistency: Discussing male infertility in a manuscript titled "Female Infertility" is inconsistent and detracts from the main focus of the work.

Thank you. We have now further reduced the mention of male infertility from the discussion section. However, as fertility is a couple issue, and resveratrol could play a role also in male fertility, we think it is important at least to mention this aspect too.

Registration in PROSPERO: The lack of registration in PROSPERO is a notable limitation, particularly given that the review is already complete. Registration is a widely recognised standard that adds transparency and credibility to systematic reviews.

Thank you for highlighting this again. We have included in the limitations section that we did not register a prior in PROSPERO.